# Low Rank Experts Enable Specialization In Dense Transformers

## Abstract

We present *StructMoE*, a drop-in augmentation for standard Transformer MLPs that improves model performance. Each MLP hosts a router that selects a token-specific top-$k$ subset from a bank of low-rank matrices. Their combined contribution is injected into the up-projection, yielding a dynamic, per-token rank-$k$ update to the base weight and executed in parallel with the up-projection via grouped GEMMs. To compare with dense baselines, we match the parameter budget of *StructMoE* by shrinking the base expansion factor to offset the router and low-rank experts' parameters. Overall FLOPS decrease because the low-rank branch is sparsely activated. StructMoE delivers token-level specialization by routing each token to structured experts inside a single dense MLP. We observe consistent quality improvements on benchmark tasks for models with upto 1.6B parameters trained on 400B tokens.

## 1 Introduction

Large language models have expanded at an extraordinary pace. Underlying this growth has been the development of the Transformer architecture (Vaswani et al., 2017) which underpins today's large language and multimodal models. Guided by empirical scaling laws (Kaplan et al., 2020) that relate loss to model size, dataset size, and compute, each new generation has pushed one or more axes by multiples, moving from hundreds of millions of parameters in early GPT variants to hundreds of billions and mixture-based trillion-parameter models(Radford et al., 2018; Brown, 2020; Smith et al., 2022; Zoph et al., 2022a; Du et al., 2022; Liu et al., 2024; Achiam et al., 2023; Comanici et al., 2025). Training runs now consume trillions of tokens, and inference increasingly targets long-context regimes with API services offering context windows on the order of $10^6$ tokens (OpenAI; Anthropic; Gemini). This growth has been enabled by steady hardware improvements, better parallelization and kernels, and data/optimization advances but these improvements have started to saturate (Hooker, 2025; Sutskever, 2024). Consequently, there is practical interest in methods that deliver more quality per FLOP while preserving the basic workload of the transformer architecture.

Within each transformer block, the position-wise feed-forward network (FFN/MLP) consumes about two-thirds of the total parameters and FLOPs because the MLP expands the hidden size $H$ to a larger $D_{\text{ff}}$ and projects back, so it owns two matrices $H \times D_{\text{ff}}$ and $D_{\text{ff}} \times H$, totaling $2HD_{\text{ff}}$ parameters whereas self-attention has four $H \times H$ projections (Q, K, V, O), i.e., $\approx 4H^2$ parameters. The FFN is uniform as every token undergoes the transformation and is dominated by large dense matrix multiplications, which are a near-ideal workload for modern accelerators. Our goal is to improve model performance at a fixed FLOP and parameter budget while preserving the GPU-friendly dense GEMMs of the MLP and adding a small token-specific computation.

With regard to preserving the dense workload of FFNs while offering specialized compute, Mixture-of-Experts (MoE) has become a popular technique which involves routing tokens to a small subset of experts to expose much larger capacity without increasing the per-token budget (Shazeer et al., 2017; Fedus et al., 2022; Du et al., 2022; Jiang et al., 2024; Zoph et al., 2022a; xAI, 2024). While MoEs achieve compelling gains over dense models, they introduce signifcant parameter sparsity and change the systems profile by introducing expert parallelism, cross-GPU communication, capacity management and routing stabilization which can complicate training and model serving.

We borrow the routing intuition from MoEs and introduce *StructMoE*, a drop-in augmentation to the standard MLP that adds a routed low-rank path inside each layer. A router scores each token

against a bank of low-rank matrices, which we call low-rank experts (LoRE), and selects a fixed top-$k$ subset whose combined contribution is injected before the up-projection's nonlinearity. All tokens follow the same computation pattern which includes one dense MLP and $k$ routed LoREs per-token.

We evaluate StructMoE in parameter-matched settings on 0.9B and 1.6B Transformers and observe improvements on language modeling benchmark tasks. In parameter-matched settings, StructMoE yields a higher accuracy of +2.20% at 0.9B and +3.99% at 1.6B (Table 2). Beyond benchmark scores, we run two ablations: (i) how to optimally split a fixed LoRE budget between the number of components L and their rank r; and (ii) how to fuse the routed path with the base MLP (additive vs. GEGLU-style gating).

## 2 RELATED WORK

Mixture-of-Experts (MoE) extends the Transformer by replacing the dense FFN with a set of $N$ experts and a learned router that maps token hidden states to expert scores. At training and inference time, a Top-$k$ gate selects a small subset of experts per token; their outputs are weighted and combined to produce the layer output. Under FLOP-matched settings, MoE models outperform dense Transformers. They leverage sparsity by activating only $k \ll N$ experts per token allowing them to access greater parameter capacity while keeping per-token compute the same as a dense model with the same FLOP budget(Shazeer et al., 2017; Fedus et al., 2022; Du et al., 2022; Jiang et al., 2024; Zoph et al., 2022a; xAI, 2024). In practice, experts are distributed across devices (expert parallelism) and tokens are shuffled via all-to-all collectives to their selected experts; routers are regularized with auxiliary load-balancing objectives (and sometimes z-loss for larger models), and capacity factors upper bound tokens per expert with potential overflow (token dropping) to maintain throughput when not using dropless MoEs. Recent work explores finer-grained experts (DeepSeek-MoE, 2024), estimation of outputs from non-activated experts (Panda et al., 2025), aux-loss free load balancing (Wang et al., 2024) along with development of optimized kernels and libraries to improve training efficiency and stability of MoEs (Gale et al., 2022; Hwang et al., 2023; Gro, 2024).

A standard formulation of MoEs includes the router as $p(x) = \mathrm{softmax}(W_r x) \in \mathbb{R}^N$ with a Top-$k$ index set $\Omega_k(x)$. Each expert $i$ is a position-wise MLP:

$$E_i(x) = (\sigma(x W_{1,i})) W_{2,i}, \qquad W_{1,i} \in \mathbb{R}^{H \times D_{\mathrm{ff}}}; W_{2,i} \in \mathbb{R}^{D_{\mathrm{ff}} \times H}, \tag{1}$$

and the MoE layer aggregates the top-$k$ expert outputs using the softmax weights as follows:

$$\mathrm{MoE}(x) = \sum_{i \in \Omega_k(x)} p_i(x), E_i(x). \tag{2}$$

**Parameter-efficient adaptation and low-rank structure.** Adapters and low-rank methods reduce fine-tuning cost by adding small trainable modules to a frozen backbone (Houlsby et al., 2019; Hu et al., 2022). LoRA models weight updates as a low-rank product added to the base matrix, yielding strong downstream performance with far fewer trainable parameters. Subsequent work explores variants such as IA$^3$, prefix/prompt tuning, and adapter fusion/ensembles (Pfeiffer et al., 2020a;b; Wang et al., 2022; Liu et al., 2022). These methods are largely *static*: the same adapters are applied to all tokens during inference. In contrast, StructMoE integrates a routed bank of low-rank components into the MLP and trains it from scratch during pretraining (no frozen backbone), so token-level specialization is learned as part of the base model rather than added post-hoc for task adaptation.

**Design of MLP blocks and gated activations.** Gated MLPs such as GLU/GEGLU/SwiGLU (Dauphin et al., 2017; Shazeer, 2020) improve the FFN by modulating the up-projection before the nonlinearity. StructMoE adopts the same principle but instantiates the modulator as a *routed, low-rank* signal: a router selects a fixed top-$k$ subset from a bank of low-rank components and adds their contribution to the base up-projection. Unlike classic gated MLPs, the modulation here is low-rank and token-dependent rather than a fixed gate shared by all tokens.

**Memory layers.** Product-Key Memory (PKM) augments a network with a very large *trainable* key–value store while keeping lookup cost small by factorizing keys into a Cartesian product of two

sub-key codebooks (Lample et al., 2019). Given a query $q \in \mathbb{R}^d$, PKM splits it as $q = [q_1; q_2]$ with $q_1, q_2 \in \mathbb{R}^{d/2}$ and maintains two codebooks $K_1 \in \mathbb{R}^{d/2 \times N_1}$ and $K_2 \in \mathbb{R}^{d/2 \times N_2}$. Each memory slot corresponds to a *product key* $(i, j)$ with score

$$s_{i,j}(q) = q_1^\top K_{1,:,i} \; + \; q_2^\top K_{2,:,j},$$

and an associated value vector $V_{i,j}$. Instead of scoring all $N_1 N_2$ keys, PKM retrieves the top-$k$ entries in $K_1$ and $K_2$, forms the $t_k \times t_k$ Cartesian product, computes $s_{i,j}$ only on this shortlist, and returns a softmax-weighted sum of the corresponding values. This gives memory capacity $|\mathcal{M}| = N_1 \times N_2$ with lookup cost proportional to $\mathcal{O}((\sqrt{|\mathcal{M}|} + k^2) \times d)$, enabling *billions* of parameters at near-constant FLOPs (Lample et al., 2019; Berges et al., 2024). In practice, the memory layer can replace an FFN in a Transformer block, providing sparse, retrieval-style capacity that complements compute-heavy dense MLPs.

**Our contribution.** StructMoE sits at the intersection of conditional computation and parameter-efficient adaptation. It adopts the *routing* principle from MoE but applies it to low-rank components localized within a dense MLP, yielding token-level specialization. Relative to static adapters, it introduces content-aware selection and does not introduce the parameter growth and the associated overhead of MoEs.

## 3 METHOD

### 3.1 MOTIVATION AND OVERVIEW

Dense Transformers allocate the same MLP computation to every token. MoEs shows that routing can between MLPs can improve model performance. Our goal is to borrow the routing intuition while maintaining most of the simplicity of a single dense MLP per layer.

**StructMoE**. Inside each MLP, we add a layer-local bank of low-rank matrices, which we refer to as low-rank experts (LoREs), and a router that selects a fixed top-$k$ subset per token; their combined contribution is injected *before* the nonlinearity of the up-projection. This yields token-level specialization with fixed per-token cost and no expert capacity or all-to-all communication. We illustrate the overall idea of StructMoE in Figure 1.

### 3.2 STRUCTMOE LAYER

Let $x \in \mathbb{R}^H$ be a token representation. A standard MLP computes

$$\text{MLP}(x) = (\sigma(xW_1))W_2, \qquad W_1 \in \mathbb{R}^{H \times D_{\text{ff}}}, ; W_2 \in \mathbb{R}^{D_{\text{ff}} \times H}, \tag{3}$$

with activation $\sigma$. StructMoE augments the up-projection with a routed low-rank update drawn from a bank of $L$ components $(A_\ell, B_\ell)\ell = 1^L$, where

$$A_\ell \in \mathbb{R}^{H \times r}, \qquad B_\ell \in \mathbb{R}^{r \times D'_{\text{ff}}}, \qquad r \ll \min(H, D'_{\text{ff}}). \tag{4}$$

A router $W_R \in \mathbb{R}^{H \times L}$ produces scores for every token followed by a softmax to obtain a distribution over the components: $s(x) = \sigma(xW_R) \in \mathbb{R}^L$. Finally, we select a fixed top-$k$ index set $\Omega_k(x)$ from $s(x)$ which introduces sparsity in the LoREs. The StructMoE up-projection is then

$$z(x) = xW'_1 + \sum_{\ell \in \Omega_k(x)} s_\ell(x) x A_\ell B_\ell, \qquad W'_1 \in \mathbb{R}^{H \times D'_{\text{ff}}} \tag{5}$$

followed by the activation and down-projection

$$\text{StructMoE}(x) = \sigma(z(x))W'_2, \qquad W'_2 \in \mathbb{R}^{D'_{\text{ff}} \times H}. \tag{6}$$

We treat $D'_{\text{ff}} < D_{\text{ff}}$ as a design knob to match a *parameter budget* with the dense baseline (§3.3). Crucially, $k$ is fixed, so every token uses the same number of LoREs, keeping FLOPs identical across the batch. The pseudocode for StructMoe can be found in Algorithm B.3.

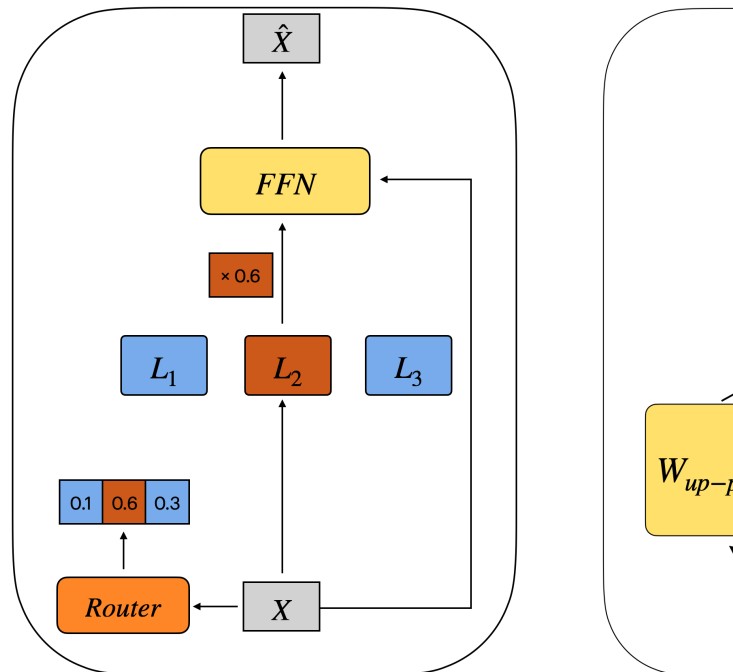
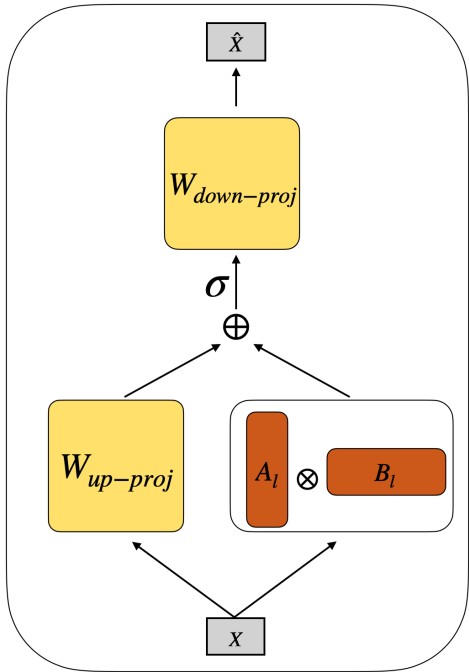

(a) Overall StructMoE layer.

(b) Inner workings in the up-projection.

Figure 1: **StructMoE: overview and inner mechanism (left to right).** *Left:* For each token $x$, a router produces scores over a bank of low-rank experts (LoRE) $\{L_i\}_{i=1}^{L}$ with $L_i \equiv (A_i, B_i)$, $\mathrm{rank}(A_i B_i) = r$. A fixed top-$k$ subset $\Omega_k(x)$ with weights $\{s_i(x)\}$ is selected, and the combined low-rank contribution $\sum_{i \in \Omega_k(x)} s_i(x)\, x A_i B_i$ is *injected before the nonlinearity* of the MLP's up-projection and then passed through the down-projection to produce the residual output $\hat{x}$. *Right:* The detailed computation inside the MLP: the dense up-projection $x W_{\text{up-proj}}$ runs in parallel with the routed low-rank path factored as $(A_{(\ell)}, B_{(\ell)})$, yielding $z = x W_{\text{up-proj}} + \sum_{\ell \in \Omega_k(x)} s_\ell(x)\, x A_{(\ell)} B_{(\ell)}$, followed by activation $z = \sigma(z)$ and the standard down-projection $\hat{x} = z W_{\text{down-proj}}$.

**LoRE-MLP interaction modes.** We consider two ways to combine the routed path with the base: (i) *Additive* (Eq. 5), and (ii) *GLU-style* gating in which the routed path modulates the activated base $\sigma(x W_1')$ multiplicatively. Empirically, we find pre-activation integration (combining before $\sigma$) to be most effective, as it lets the routed path influence the activation's activation dynamics.

### 3.3 PARAMETER AND COMPUTE ACCOUNTING

We match the dense baseline's parameter budget and analyze FLOPs to ensure a fair comparison.

**Parameters.** Ignoring biases, a dense MLP has $P_{\text{dense}} = 2 H D_{\text{ff}}$ parameters. The StructMoE layer has

$$P_{\text{StructMoE}} = 2 H D_{\text{ff}}' + Lr(H + D_{\text{ff}}') + HL \tag{7}$$

parameters where the last term accounts for the router $W_R$ computation. Solving $P_{\text{StructMoE}} = P_{\text{dense}}$ for $D_{\text{ff}}'$ gives

$$D_{\text{ff}}' = \frac{2 H D_{\text{ff}} - HL(r+1)}{2H + Lr}. \tag{8}$$

This trades a reduction in the base expansion for the router and LoRE parameters.

**FLOPs.** Counting multiply–adds as 2 FLOPs, the dense MLP costs $F_{\text{dense}} = 4 H D_{\text{ff}}$ per token. StructMoE costs

$$F_{\text{StructMoE}} = 4 H D_{\text{ff}}' + 2HL + 2kr(H + D_{\text{ff}}'), \tag{9}$$

covering the base MLP, the router's matrix-vector multiplication, and $k$ LoREs. Substituting Eq. 8, the ratio $\rho = F_{\text{StructMoE}}/F_{\text{dense}} < 1$ as $k < L$ and $(A_\ell, B_\ell)$ are always low rank.

## 3.4 EXECUTION: GROUPED GEMMs

Evaluating $\sum_{\ell \in \Omega_k(x)} s_\ell(x) x A_\ell B_\ell$ efficiently is the key systems question. We execute the low-rank path with *Grouped GEMM* operations.

1. **Token grouping.** For a batch of tokens, we compute $\Omega_k(x)$ and gather tokens by selected component index to form contiguous mini-batches per $\ell$.

2. **First low-rank multiply.** For each active $\ell$, compute $X_\ell A_\ell$ where $X_\ell$ stacks the grouped tokens; this produces shape $(n_\ell \times r)$.

3. **Second multiply.** Multiply by $B_\ell$ to obtain $(n_\ell \times D'_{\text{ff}})$ and scatter back to token positions with weights $\beta\ell(x)$.

4. **Fuse with base.** Add to $xW'_1$ and proceed with activation and $W'_2$.

This pattern avoids materializing $A_\ell B_\ell$.

## 3.5 DESIGN CHOICES AND INTUITION

**Why routed low rank?** Our motivation follows the MoE intuition: when different parts of the data distribution are handled by specialized mappings, quality improves at similar per-token compute. StructMoE aims to capture a *lightweight* version of that specialization *inside* a dense MLP, without introducing separate experts. Concretely, we view the up-projection as a shared base plus a token-conditioned, low-rank adjustment, The bank $\{A_\ell B_\ell\}_{\ell=1}^L$ acts like a set of small "low-rank experts," and the router chooses a fixed top-$k$ of them per token. Injecting this adjustment *before* the nonlinearity modulates which features are amplified by the activation, while keeping the workload as dense GEMMs and the per-token FLOPs fixed. We do not claim full equivalence to MoE; rather, this is a low-rank approximation to content-dependent specialization that trades a modest reduction in base expansion $D'_{\text{ff}}$ for a routed bank of low-rank updates. Our ablations (Sec. B.1, B.2) inform practical settings of $(r, L, k)$ and the fusion choice at the scales we study.

**Pre-activation integration.** Injecting the routed path before $\phi$ lets it interact with the activation's gating, which we find more effective than post-activation addition.

**Where to attach LoREs.** In our main experiments, we only attach LoREs to the up-projection. We experimented with attaching them to the down-projection as well but we found that it offers a weaker compute–quality trade-off and introduces additional overhead.

## 3.6 TRAINING DETAILS

**Routing, fixed $k$, and auxiliary losses.** Fixing $k$ keeps per-token compute identical across the batch. Without additional regularization, however, the router can *collapse*—concentrating most tokens on a small subset of LoREs while leaving others rarely selected—which wastes parameters, reduces specialization/diversity, and creates tiny per-LoRE micro-batches that hurt kernel efficiency. To prevent collapse, we add a Switch-style load-balancing loss on the router's probabilities (Fedus et al., 2022) with a coefficient of 0.01. This loss encourages the routing scores to remain close to a uniform target. In practice, this maintains *relatively balanced* utilization across the $L$ LoREs (Sec. 4.5; Fig. 4). We initially experimented with a z-loss (Zoph et al., 2022b) on router logits to discourage large values, but it had a negligible impact on performance and thus we did not use it for our actual experiments.

**Initialization.** We use *SmallInit* (Nguyen & Salazar, 2019) for $W'_1$, $A_\ell$, $B_\ell$, and $W_R$: weights are drawn $\mathcal{N}(0, \sigma^2)$ with $\sigma = \sqrt{2/(5d)}$ for input dimension $d$. For $W'_2$ we use the *DeepNorm* (Wang et al.) initializer: $\mathcal{N}(0, \sigma^2)$ with $\sigma = 2/(N_{\text{layers}}\sqrt{d})$.

**Hyperparameters.** We tuned $L$ (number of LoREs), $r$ (rank), and $k$ (active per token) under a fixed parameter budget using Eq. 8. Empirically, we obtain the best results when $k = 1$, $r = 16$, and $L = 16$. Section B.1 contains more details regarding this choice.

| Variant | Layers | Hidden $H$ | FFN Dim | Num LoREs | LoRe Rank | Parameters | Tokens |
|---------|--------|-----------|---------|-----------|-----------|-----------|--------|
| Dense | 8 | 2048 | $D_{\text{ff}} = 7168$ | - | - | 0.9B | 345B |
| StructMoE | 8 | 2048 | $D'_{\text{ff}} = 6618$ | 16 | 16 | 0.9B | 345B |
| Dense | 24 | 2048 | $D_{\text{ff}} = 7168$ | - | - | 1.6B | 400B |
| StructMoE | 24 | 2048 | $D'_{\text{ff}} = 6618$ | 16 | 16 | 1.6B | 400B |

Table 1: **Model sizes and parameter matching.** We evaluate two decoder-only Transformers at 0.9B and 1.6B parameters. StructMoE matches the dense parameter budget at each scale by reducing the MLP expansion from $D_{\text{ff}}{=}7168$ to $D'_{\text{ff}}{=}6618$ while adding the routed LoREs.

### 3.7 COMPLEXITY AND MEMORY

Per token, StructMoE adds a router cost $HL$ and $k$ low-rank matrices $(kr\,(H{+}D'_{\text{ff}}))$ on top of the base MLP cost $(2HD'_{\text{ff}})$. We reduce $D'_{\text{ff}}$ to keep the total parameter budget matched.

*Practical overhead.* StructMoE adds a small set of extra steps around the base MLP: we bin tokens by their selected LoRE, gather them into *per-LoRE batches*, run the two low-rank GEMMs per batch, and then scatter results back. Let $T$ be the number of tokens in the layer, $k$ the router's top-$k$ and $n_\ell$ as the per-LoRE batch size. Aggregating across all LoREs, the low-rank intermediates and outputs are tensors of shape $(Tk){\times}r$ and $(Tk){\times}D'_{\text{ff}}$ before we scatter back to the original token order.

*Memory footprint.* We need to store the LoREs $\{A_\ell, B_\ell\}_{\ell=1}^{L}$ and the router $W_R$, plus (i) top-$k$ routing tensors of size $Tk$, (ii) the gathered *per-LoRE batches* totaling $THk$ activations in feature space, and (iii) the low-rank intermediates/outputs kept as tensors $(Tk){\times}r$ and $(Tk){\times}D'_{\text{ff}}$ prior to the scatter. In our setting ($k = 1$, $r = 16$, $L = 16$), this leads to a small increase in activations. In all our runs we were able to keep the same per-GPU batch size and sequence length as the dense baseline. The main runtime cost remains the extra gather/scatter traffic and the launches for many small grouped GEMMs. See Sec. 4.6 for more details.

## 4 EVALUATION

### 4.1 MODELS

We evaluate two decoder-only Transformer sizes—900M and 1.6B—each with a parameter-matched Dense and StructMoE variant (Table 1). The 900M models use 8 layers; the 1.6B models use 24 layers. Across all variants we hold the backbone fixed: hidden size $H{=}2048$, 32 attention heads, rotary position embeddings (Su et al., 2024) and LayerNorm. We use the Llama3 tokenizer (3, 2024) for both models without weight-tying resulting in approximately 525M embedding parameters.

StructMoE matches the dense parameter budget at each scale by reducing the MLP expansion from $D_{\text{ff}}{=}7168$ to $D'_{\text{ff}}{=}6618$ while adding the routed low-rank bank (see Eq. 7 in §3.3); all other architectural choices are identical between Dense and StructMoE. Training settings are also shared. We train with a sequence length of 2048 and a global batch size of $2^{21}$ tokens per step.

### 4.2 DATA

Pretraining uses the **FineWeb-Edu** dataset (Penedo et al., 2024), an education-oriented subset of FineWeb with heuristic quality filters and deduplication which we tokenize using Llama-3 Tokenizer 3 (2024).

### 4.3 TRAINING SETUP

**Implementation.** We train with *Megatron-LM* (Shoeybi et al., 2019) and Deepspeed Aminabadi et al. (2022), integrating StructMoE's routed LoREs using grouped-GEMM kernels provided in *Megablocks*. (Gale et al., 2022). Routing uses Top-$k$ ($k = 1$) with a load-balancing coefficient of 0.01.

**Hardware and parallelism.** We used 32 NVIDIA A100 40GB GPUs for training, connected via AWS EFA. The setup consisted of 4 nodes with 8 GPUs per node. Our models fit on a single GPU, so we disable tensor and pipeline parallelism and use data parallelism (DP) only. We use the DeepSpeed ZeRO Stage 1 optimizer (Rajbhandari et al., 2020).

**Optimization and schedules.** For optimization and schedules, we use Adam with a base learning rate of $3 \times 10^{-4}$ and a minimum learning rate of $3 \times 10^{-5}$. The learning rate follows a cosine decay schedule with a linear warm-up of 0.1%.

## 4.4 RESULTS

| Benchmark | 0.9B Parameter Transformer | | | 1.6B Parameter Transformer | | |
|---|---|---|---|---|---|---|
| | Score | | | Score | | |
| | Baseline | StructMoE | Diff (%) | Baseline | StructMoE | Diff (%) |
| ARC-Challenge | 31.8 | 33.2 | +4.40% | 34.6 | 36.6 | +5.78% |
| ARC-Easy | 64.4 | 66.6 | +3.42% | 70.5 | 72.1 | +2.27% |
| BoolQ | 61.7 | 61.9 | +0.32% | 62.8 | 64.1 | +2.07% |
| HellaSwag | 38.3 | 38.6 | +0.78% | 57.4 | 59.6 | +3.83% |
| Lambada | 35.2 | 37.9 | +7.67% | 45.5 | 48.5 | +6.59% |
| MNLI | 35.3 | 35.9 | +1.70% | 36.4 | 37.8 | +3.84% |
| OpenBookQA | 26.0 | 25.0 | -3.85% | 27.0 | 28.2 | +4.44% |
| PubMedQA | 53.6 | 56.4 | +5.22% | 55.6 | 60.0 | +7.91% |
| SciQ | 77.7 | 79.7 | +2.57% | 83.6 | 86.7 | +3.71% |
| TruthfulQA | 23.0 | 23.1 | +0.43% | 19.2 | 20.9 | +8.85% |
| Winogrande | 53.1 | 52.8 | -0.56% | 58.0 | 59.8 | +3.10% |
| Average | 45.5 | 46.5 | +2.20% | 50.1 | 52.1 | +3.99% |

Table 2: The table compares StructMoE against parameter-matched dense baselines for two decoder-only Transformers (0.9B and 1.6B) on standard downstream benchmark tasks (higher is better). All models are trained under identical data and optimization settings. At 0.9B, StructMoE improves over the dense baseline by **2.20%**. At 1.6B, the gain increases to **3.99%**.

Table 2 reports the benchmark scores for StructMoE and the parameter-matched dense baselines at both scales. At 0.9B, StructMoE improves over the dense baseline by **2.20%**. At 1.6B, the gain increases to **3.99%**. StructMoE outperforms the dense baselines in the majority of cases at both model sizes, suggesting that the routed low-rank path provides useful token-specific specialization.

Figure 2 compares the token-scaled training loss for the parameter-matched dense and StructMoE models at 1.6B (Fig. 2a) and 0.9B (Fig. 2b). All runs use the same backbone (hidden size 2048, 32 heads, sequence length 2048) and optimizer/schedule; the only architectural change is replacing the dense MLP with StructMoE while reducing $D'_{\text{ff}}$ to match parameters. These curves mirror the benchmark gains reported in Table 2: relative improvements over the dense baseline are higher for the 1.6B model.

## 4.5 ROUTING ANALYSIS ACROSS LoREs

To assess utilization and check for router collapse, we measure the fraction of tokens for which each low-rank expert (LoRE) is selected by the router. For every layer, we count how often a LoRE appears in the token's top-$k$ set (with $k \in \{1\}$ in our runs), normalize by the total number of token–selections at that layer, and plot the distribution across depth. As shown in Fig. 4, routing is relatively balanced among LoREs within each layer indicating that router collapse does not oc-

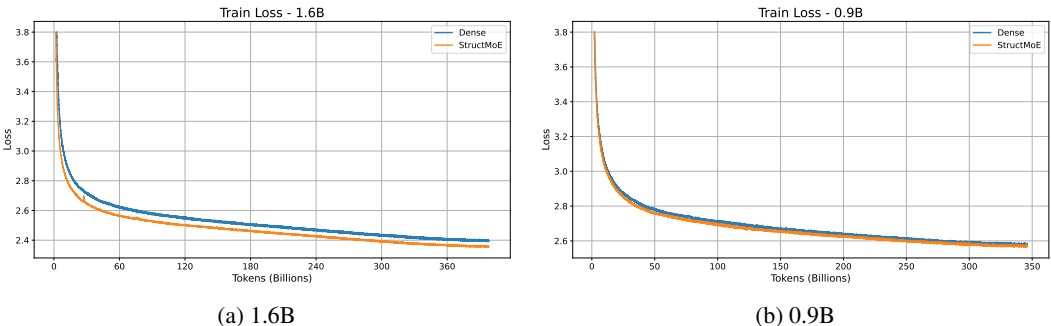

(a) 1.6B

(b) 0.9B

Figure 2: **Training curves (Dense vs. StructMoE) at two scales.** Loss vs. tokens for parameter-matched models.

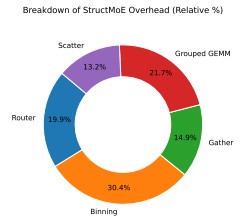

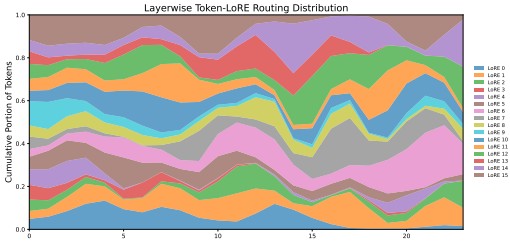

Figure 3: **Breakdown of StructMoE overhead** Most of the overhead comes from routing/packing and memory movement rather than the low-rank multiplies themselves.

Figure 4: **Layerwise token–LoRE routing distribution.** For each Transformer layer (x-axis), the stacked areas show the share of token selections assigned to each low-rank expert (LoRE) after top-$k$ routing (y-axis sums to 1 per layer). We observe relatively balanced utilization of LoREs across model layers.

cur. This balance is consistent with the Switch-style load-balancing loss used during training and indicates that StructMoE leverages the full bank of LoREs rather than over-relying on a few.

## 4.6 STRUCTMOE OVERHEAD

StructMoE introduces a measurable runtime overhead: at 0.9B it increases iteration time by **6.53%** and reduces achieved TFLOPs by **4.23%**; at 1.6B, the overhead is **5.77%** in iteration time and a **6.52%** reduction in TFLOPs (Table 3). To understand where this comes from, we break the incremental cost into five parts: *routing*, *binning*, *gather*, *grouped GEMM*, and *scatter*. Figure 3 shows their relative shares. The striking observation is that the majority of overhead is not the matrix-multiplications for the LoRE computations but the orchestration and data movement around them.

**Why this overhead appears.** (i) *Router:* a per-token matrix multiplication $xW_r$ and Top-$k$ selection are small, but they run every layer. (ii) *Binning:* building token groups for each selected LoRE requires sorting/histograms and per-LoRe token counts. (iii) *Gather/Scatter:* moving tokens into per-LoRE batches and then restoring the original order adds extra reads/writes and memory traffic. (iv) *Grouped GEMM:* we compute the low-rank updates as Grouped GEMMs over the per-LoRE batches. We borrowed this approach from modern MoE implementations.

| Scale | Avg. Iter time (ms/step) | | | Avg. TFLOPs | | |
|-------|-------|-----------|----------|-------|-----------|----------|
|       | Dense | StructMoE | Diff (%) | Dense | StructMoE | Diff (%) |
| 0.9B  | 1.99  | 2.12      | +6.53    | 197   | 189       | −4.23    |
| 1.6B  | 4.85  | 5.13      | +5.77    | 196   | 184       | −6.52    |

Table 3: **Runtime and throughput at two scales, with overhead.** Per-step wall-clock time and achieved TFLOPs for parameter-matched Dense vs. StructMoE models.

**Design implications.** Since the overhead per layer is dominated by fixed orchestration rather than the LoRE arithmetic, the most effective way to amortize it is to increase per-layer compute. The added cost is paid once per layer (routing, binning, gather/scatter) and thus deeper models incur it more times. For a fixed parameter/compute budget, it is therefore better to shift capacity into **wider** layers (larger $H$ and $D'_{\mathrm{ff}}$) and use **fewer** layers, so we pay the overhead fewer times and the grouped GEMMs are larger and more arithmetically intense. Additionally, this overhead becomes smaller as model size increases (by way of wider models) as the time spent in the standard transformer operations dominates the overall computation in the model. Consequently, the LoRE orchestration becomes a *smaller fraction* of total step time.

## 5 LIMITATIONS

While StructMoE improves quality at matched parameters and lower theoretical FLOPs, our study has several limitations.

**Scale of evaluation.** We report results at 0.9B and 1.6B parameters (Sec. 4.4). The observed gains grow across these two scales, but we have not yet validated the trend beyond 1.6B parameter models. Larger models (e.g., 7B) may exhibit different interactions between routing, LoREs and the MLP.

**Dense-Transformer–only evaluation.** We restrict our study to *dense* Transformers—one MLP per layer augmented with routed low-rank components—so results are directly comparable to a dense baseline. We do not stack StructMoE on top of a traditional MoE, and it is unclear how two kinds of sparsity/routing would interact. Combining full experts with in-expert low-rank experts could unlock finer specialization and provide many more routing combinations offering a potentially better scaling path than simply adding more full experts. However, it will also compound overheads (two routers, extra gathers/scatters etc). We leave the exploration of this to future work.

**Implementation overhead masks some practical gains.** The current `megablocks` grouped-GEMM path adds real routing/packing costs (binning, gather/scatter, many small-group matmuls) that reduce achieved TFLOP/s versus a plain dense MLP masks practical gains. We believe that improvements in MoE style kernels should help trim the overhead.

## 6 CONCLUSION

We introduced **StructMoE**, a drop-in modification to dense Transformers that routes a fixed number of low-rank components inside the MLP, enabling token-level specialization. In parameter-matched settings, StructMoE improves on benchmark scores over dense baselines at two scales: **+2.20%** (0.9B) and **+3.99%** (1.6B) and shows balanced routing across components. Ablations indicate that a mid-rank/mid-count setting ($r=16$, $L=16$) is most effective and that additive and GEGLU fusions converge to similar final loss.

On the implementation side, the current grouped-GEMM implementation introduces nontrivial routing/packing overhead that reduces achieved TFLOPs compared to the dense baseline. Fewer but wider layers help amortize this cost, and kernel/library improvements should further trim it.

# 7 ETHICS

This work focuses on methods for improved pretraining techniques for large language models through routed low rank experts. As such, this work does not raise novel ethical or societal risks beyond those already associated with large language models.

# 8 REPRODUCIBILITY

We have made significant the following efforts to ensure the reproducibility of our results.

1. **Model details** Details for our models are included in Section 4.1

2. **Dataset.** The dataset is described in Section 4.2.

3. **Hyperparameters & training.** Optimizer, training schedule, batch sizes, hardware setup and experimental details are in Section 4.3.

4. **Code.** We use publicly available libraries for training (details in 4.3 and we will release our code upon acceptance.

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

## A  USE OF LLMS

We used LLMs for plotting code, latex formatting and finding relevant work.

## B  ABLATIONS

### B.1  CHOOSING RANK AND NUMBER OF LORES

We study how to allocate the LoRE budget between *rank* ($r$) and *count* ($L$). In a small-scale setting, we sweep four configurations while keeping the total LoRE budget and overall parameters constant

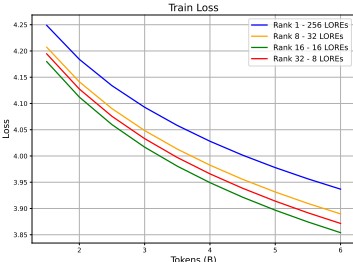 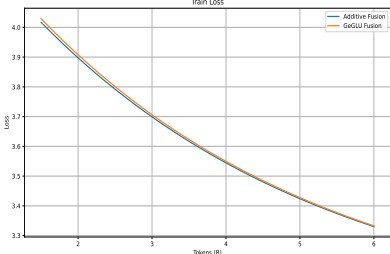

Figure 5: **LoRE rank–count sweep at constant budget.** Training loss vs. total training tokens for four StructMoE configurations with fixed LoRE budget $L \times r = 256$ and matched overall parameters. Curves correspond to $r{=}1, L{=}256$, $r{=}8, L{=}32$, $r{=}16, L{=}16$, and $r{=}32, L{=}8$.

Figure 6: **Additive vs. GEGLU fusion for StructMoE.** Training loss (y-axis) vs. seen tokens in billions (x-axis) with identical $(k, r, L)$ and parameter budgets. Both fusion strategies converge to the same final loss, but *GEGLU* starts with a higher initial loss.

by fixing $L \times r = 256$ and adjusting $D'_{\text{ff}}$ via Eq. (8). The four settings are: $(r{=}1, L{=}256)$, $(r{=}8, L{=}32)$, $(r{=}16, L{=}16)$, and $(r{=}32, L{=}8)$. Figure 5 plots training loss vs. tokens for these variants. Empirically, $r{=}16$, $L{=}16$ provides the best trade-off, consistently achieving the lowest loss at matched tokens. Therefore, we use $(r{=}16, L{=}16)$ in all our large-scale experiments.

## B.2 ADDITIVE VS. GEGLU FUSION

We compare two ways of combining the routed LoRE path with the base up-projection: *additive* (pre-activation sum) and GeGLU fusion (the routed path modulates the activated base). Keeping $(k, r, L)$, $D'_{\text{ff}}$, and all optimization settings identical, both variants converge to essentially the same training loss at matched tokens. However, as shown in Fig. 6, GeGLU exhibits a higher initial loss compared to additive fusion. We used the additive fusion for all our large-scale experiments.

## B.3 STRUCTMOE PSEUDOCODE

---

**Algorithm 1** StructMoE MLP forward (per layer)

---

**Require:** Input Batch $X \in \mathbb{R}^{T \times H}$, weights $W'_1, W'_2, W_R, (A_\ell, B_\ell)\ell = 1^L$, top-$k$, activation $\phi$, softmax $\sigma$
1: $s(X) \leftarrow \sigma(X W_R)$           (router scores)
2: For each token $t$, select $\Omega_k(X_t)$ and weights $s_\ell(X_t)$
3: Group tokens by selected index $\ell$; form mini-batches $X_\ell$
4: $Z_0 \leftarrow X W'_1$           (base up-projection)
5: **for** each active $\ell$ **do**
6:     $R_\ell \leftarrow (X_\ell A_\ell) B_\ell$           (grouped GEMMs)
7:     Scatter $s_\ell(\cdot) R_\ell$ back into token positions
8: **end for**
9: $Z \leftarrow Z_0 + \sum_\ell \text{scatter}(s_\ell R_\ell)$
10: $Y \leftarrow \phi(Z) W'_2$ **return** $Y$

---

