# OpenReview forum: "Low Rank Experts Enable Specialization In Dense Transformers"
_ICLR.cc/2026/Conference — ICLR 2026 Conference Withdrawn Submission_

### Official Review · Reviewer_175z · 2025-10-26

**Soundness:** 2
**Presentation:** 2
**Contribution:** 2
**Rating:** 4
**Confidence:** 3

**Summary:**

The paper proposes an alternative/improvement to traditional dense transformer MLPs that augments the up-projection inside each layer with a routed low-rank expert (LoRE). The authors combine the concept of low-rank approximation with expert routing to introduce multiple low-rank approximations that act as experts within each dense MLP. They motivate LoRE with practical difficulties around MoE models and the compute-inefficiency of dense models. They seek to improve model performance at a fixed FLOP and parameter budget while preserving the GPU-friendly dense GEMMs of the MLP. The authors show the improvement through StructMoE over dense baselines across many benchmarks, and analyze the routing balance across different layers.

**Strengths:**

1. The authors provide a good presentation of related work and common practices in training MoE models.
2. The authors provide a good foundation for LoRE formulations, keeping in mind the engineering and practical considerations for model training (leveraging GPU-friendly dense GEMMs).
3. The authors provide clear details on parameter and compute calculations, and are very transparent in their runtime overhead and limitations.
4. The authors make clear that the design choices related to LoRE (placement at up-projection, pre-activation integration) are empirically well motivated, please provide the exact numbers in the appendix.
5. The authors benchmark their proposed method against a dense baseline across many task domains, displaying significant improvements in scores across most benchmarks.

**Weaknesses:**

- The paper is missing evaluations of test loss/perplexity on a general corpus like FineWeb. These evaluations are standard for language models.
- Despite the sparsity, StructMoE is actually slower than the parameter-matched dense baseline.
- It’s unclear how the performance of StructMoE compares with FLOP/Parameter-matched MoE models.

*Low level comments:*

1. Typo line 20 “upto”
2. Figure 1 can be placed on the second page at the top to provide more context and better framing of the proposed approach throughout the paper.
3. Paragraph titles are inconsistent in sections 3.5, 3.7, 4.6

**Questions:**

See weaknesses.

---

### Official Review · Reviewer_w376 · 2025-10-27

**Soundness:** 1
**Presentation:** 3
**Contribution:** 1
**Rating:** 2
**Confidence:** 5

**Summary:**

This paper proposes StructMoE, which introduces low-rank parameters into the up-projection of the MLP. These low-rank parameters are selected by a router, similar to how a standard MoE operates.

**Strengths:**

The writing is clear, and the experimental details provided are sufficient.

The discussions and analyses are diverse.

**Weaknesses:**

1. I have concern about the long-term stability and sustainability of the method’s performance advantage. As shown in Figure 2(a), two loss spikes occur around the 30B and 70B token checkpoints; after each spike, the loss gap between the proposed method (orange line) and the dense model (blue line) shrinks. This trend is also observed in Figure 2(b), where the performance gap continues to narrow as training progresses. To sum up, there is a risk that this method only accelerates convergence and is effective only in the early stages of pre-training.

2. I am afraid I do not agree with the caption of Figure 4, where you state that the utilization of LoREs is "relatively balanced." It is obvious that LoRE13 is barely used and even "dies out" in deep layers. LoRE6 and LoRE4 account for almost half of the tokens, which is clearly not balanced at all.

3. The motivation seems problematic (Section 3.1). You mention wanting to "borrow the routing intuition while maintaining most of the simplicity of a single dense MLP per layer." However, you actually introduce routing complexity similar to that of MoE. The implementation complexity of StructMoE is no simpler than that of a vanilla MoE. If it is not simpler and more effective, why would researchers use StructMoE instead of MoE? In fact, I noticed that you used GEMM kernels in megablocks, which confirms that the complexity of StructMoE is comparable to that of a vanilla MoE.

4. If your motivation is to enable dense MLPs to benefit from MoE’s core ideas, and if you argue that your method is simpler, the most convincing evidence would be a direct comparison with MoE. Since I do not see how your method is simpler than MoE, it would be better to demonstrate that it has a performance advantage over MoE.

5. The "up-proj-then-down-proj" MLP structure is rarely used in modern LLMs. How would you adapt your method to the widely used SwiGLU MLP? Would it still be effective?

6. There are numerous papers on combining LoRA and MoE, such as https://arxiv.org/pdf/2312.09979 and https://aclanthology.org/2025.findings-naacl.284.pdf. In terms of technique and novelty, this paper does not stand out.

7. The abstract in the openreview system is inconsistent with that in the submission file.

**Questions:**

See weaknesses.

---

### Official Review · Reviewer_uhrQ · 2025-10-29

**Soundness:** 2
**Presentation:** 2
**Contribution:** 1
**Rating:** 2
**Confidence:** 3

**Summary:**

The paper contribution is a hybrid architecture that augments a dense FFN with a low-rank MoE. However, the concept of integrating the MoEs into the transformer block is not new. Using routing to send tokens to different places, is not new. Several works have explored similar ideas to get the benefits of MoE over the FFN layers.

**Strengths:**

While the paper proposes StructMoE as a novel solution, its contributions are not differentiated from the existing works.

**Weaknesses:**

Augmenting FFNs with a routed path is not a new concept. The paper fails to discuss or differentiate this method from other hybrid MoE architectures that also aim to combine the benefits of dense pathways with specialized experts. The idea of routing to low-rank experts is also not new. For example, LoRAMoE (Dou et al., 2024) has already used mixing MoE with LoRA adapters for efficient fine-tuning. The method described in section 3.3 for matching parameter counts (reducing $D'_{ff}$ to make room for the LoREs) is a standard, and not novel. The paper's efficiency also are unclear.

**Questions:**

Based on my knowledge, in standard MoE, the L experts are sharded across L GPUs. So, the layer-local design described here, implies that all L experts must be stored on every GPU, as any token could be routed to any of them. This means the memory per device can be P_{dense} + P_{router} + P_{L experts}, which is much larger than the dense baseline it's parameter-matched (especially if the baseline's parameters can be sharded). So, this makes difficult to understand the model's efficiency.

Additionally, the authors didnt  provide details on router training, stability, or load balancing, which are the sources of complexity in all MoE models. The FLOP calculation (Eq. 9) is not clear or maybe misleading. Please provide actual throughput numbers (e.g., training steps/second or tokens/second) compared to the dense baseline.
 Related to this, at the end of Section 3.3, the paper simply concludes the FLOPs ratio is $\rho < 1$ without a clear proof for this inequality.

Are all L LoREs replicated on every single GPU? If so, how this can be a parameter-matched comparison when the memory-per-GPU is significantly higher?

Section 3.2 says: "A router ... produces scores for every token followed by a softmax to obtain a distribution ...".
But the equation given by $s(x) = \sigma(xW_R)$. Generally, the $\sigma$ is used for activations like Sigmoid or ReLU, not Softmax.

---

### Official Review · Reviewer_cc7q · 2025-11-01

**Soundness:** 3
**Presentation:** 2
**Contribution:** 2
**Rating:** 4
**Confidence:** 3

**Summary:**

This paper proposes StructMoE, a drop-in augmentation to standard Transformer MLPs that introduces a lightweight form of token-level specialization without changing the dense compute pattern of transformers. Instead of replacing the FFN with large sparse experts (as in traditional MoE models), StructMoE adds a bank of low-rank experts within each MLP layer. A small router selects a fixed top-k subset of these low-rank components per token, and their combined contribution is injected before the activation in the up-projection. Experiments on 0.9B and 1.6B parameter language models trained on 345B–400B tokens show consistent gains (+2.2% and +3.99%) over parameter-matched dense baselines across a suite of benchmarks (ARC, BoolQ, SciQ, etc.).

**Strengths:**

1. Interesting yet simple idea. The work cleverly integrates the advantages of conditional computation and low-rank adaptation. It embeds the ideas of MoE and LoRA into MLP blocks.
2. Strong empirical evaluation. Results are presented on large-scale models (up to 1.6B parameters) with realistic training budgets and consistent benchmark improvements across multiple NLP tasks.

**Weaknesses:**

1. The method introduces ~5–7% slower iteration times and reduced TFLOPs efficiency, mainly due to routing and gather/scatter operations, partially negating the FLOP advantage. Can you analyze the inference time and training time in more detail? These numerous routers would significantly increase communication overhead, making it difficult for industrial adoption.
2. Such a design doesn't seem to address the core issues of the Transformer (e.g., long-context modeling and memory). The performance gains come at the cost of efficiency, making it unlikely to be widely adopted in the industry.

**Questions:**

1. How sensitive is StructMoE’s performance to the choice of rank (r), number of experts (L), and top-k?
2. Are the observed 2–4% average improvements statistically significant, given the size of benchmark variance?

---

### Note · Authors · 2025-12-01

I have read and agree with the venue's withdrawal policy on behalf of myself and my co-authors.